# Multimodal Relation Extraction via a Mixture of Hierarchical Visual Context Learners

## ABSTRACT

Multimodal relation extraction is a fundamental task of multimodal information extraction. Recent studies have shown promising results by integrating hierarchical visual features from local regions, like image patches, to the broader global regions that form the entire image. However, research to date has largely ignored the understanding of how hierarchical visual semantics are represented and the characteristics that can benefit relation extraction. To bridge this gap, we propose a novel two-stage hierarchical visual context fusion transformer incorporating the mixture of multimodal experts framework to effectively represent and integrate hierarchical visual features into textual semantic representations. In addition, we introduce the concept of hierarchical tracking maps to facilitate the understanding of the intrinsic mechanisms of image information processing involved in multimodal models. We thoroughly investigate the implications of hierarchical visual contexts through four dimensions: performance evaluation, the nature of auxiliary visual information, the patterns observed in the image encoding hierarchy, and the significance of various visual encoding levels. Empirical studies show that our approach achieves new state-of-the-art performance on the MNRE dataset.[1]

## CCS CONCEPTS

• **Information systems → Multimedia and multimodal retrieval**; • **Computing methodologies → Information extraction**.

## KEYWORDS

Multimodal Relation Extraction, Multimodal Fusion

**ACM Reference Format:**
Anonymous Author(s). 2024. Multimodal Relation Extraction via a Mixture of Hierarchical Visual Context Learners. In *Proceedings of the ACM Web Conference 2024 (WWW '24)*. ACM, New York, NY, USA, 12 pages. https://doi.org/XXXXXXX.XXXXXXX

## 1 INTRODUCTION

Relation extraction (RE) [24, 41] aims at identifying the semantic relationships between entities in the text. This area has gained traction in research due to its role as a core subtask in many web applications that require relational understanding, including web

---

[1]The source code: https://anonymous.4open.science/r/HVFormer-9D85

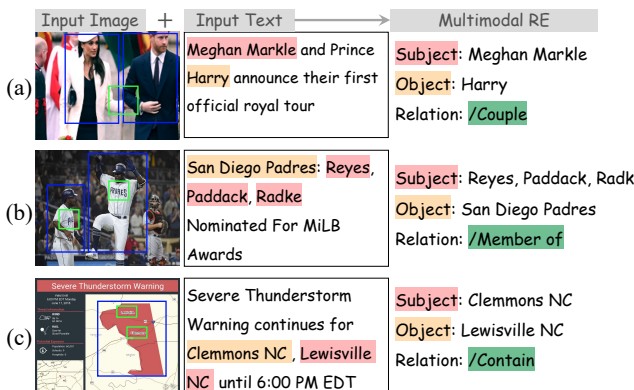

**Figure 1: The multi-scaled image regions retain significant context for extracting relations. The relative local image regions are bounded by green boxes, while the image regions with more global information are enclosed in blue boxes.**

mining [18], question answering [17], and information retrieval [13]. Meanwhile, the rapid growth of social media platforms like Twitter has resulted in an immense volume of user-generated content that spans multiple modalities, including text and images. This has led to a growing interest in multimodal relation extraction (MRE) [45] as a critical research direction. The rationale behind this is that while text might sometimes lack sufficient context information (e.g., instances of short texts), images may provide additional contextual cues to aid in decoding the relation.

The main challenge of MRE is how to integrate image information into the text representation learning process to improve relation extraction. Previous work primarily leverages pre-trained encoders for extracting visual features, broadly classified into two categories: i) Graph alignment-based methods [39, 44] construct textual and visual graphs from text-image pairs. These methods then adopt graph alignment algorithms to map visual clues with textual content to learn multimodal representations. ii) Representation learning-based methods [32, 43] directly map the semantics of texts and images in vector spaces, and develop methods to combine text and image representations to obtain multimodal representations. Both methodologies use the learned multimodal representations to extract relations. Representation-based methods have been shown to deliver superior performance than graph alignment-based models in previous research [4, 5]. This discrepancy can be attributed to the possibility of discarding some original image features during visual graph construction.

Although previous representation-based methods have achieved notable performance [4], these methods primarily focus on leveraging the high-level features obtained from the output of pre-trained visual encoders such as ResNet [10] and Vision Transformer (ViT) [8], neglecting the impact of utilizing the hierarchical visual

context information. For example, in Figure 1a, we can recognize the two people in the human region of the image as *Prince Harry* and *Meghan Markle*. The relationship between them, the *couple*, can be inferred from the local image region that depicts the linked arms. Similar scenarios are depicted in Figure 1b and c. Hence, it is imperative to leverage the hierarchical visual contexts, ranging from local image regions to global regions that make up the entire image, to benefit multimodal RE. Recent models [4, 43] implicitly exploit hierarchical visual features by applying modality alignments at the same layer of unimodal encoders. Chen et al. [5] consider the significance of different ResNet [10] blocks to fuse visual features. Nevertheless, existing research has yet to comprehend how hierarchical visual semantics are represented and what hierarchical characteristics can benefit the MRE task.

Due to the recent success of vision transformers in computer vision [8, 33] and multimodal information extraction [4, 35], the fine-grained hierarchical visual features learned by ViT can be utilized to enhance multimodal RE. Although each transformer block in ViT follows the same procedure to aggregate the entire image information, recent research [14] suggests that visual attention heads in earlier layers tend to focus on local and sparse regions, while most heads in deeper layers globally attend to regions scattered across the image. Consequently, different ViT layers can encode visual context on different levels and directly using the high-level image embeddings of the vision encoder as prior methods [12, 35, 38], may be suboptimal for extracting relations. As illustrated in Figure 1b, the visual information in the critical logo region may be diluted by aggregating information from other areas of the image.

In this paper, our aim is to delve into the inherent mechanisms of multimodal models that employ vision transformers as image decoders for learning hierarchical visual contexts and to ascertain their implications on the task of multimodal relation extraction. To achieve this, we propose a novel **H**ierarchical **V**isual context **F**usion transformer (**HVFormer**), which incorporates the mixture of multimodal experts (MoME) framework to adaptively represent and integrate hierarchical visual features into textual semantic representations. In each vision transformer layer, a specific expert network is implemented with the textual-guided visual attention block (TVA) for the intra-level aggregation of token-aware visual semantic representations. An expert router is utilized to predict the importance coefficients of experts, facilitating the inter-level aggregation of multimodal representations. To gain a deeper understanding of the intrinsic mechanisms behind hierarchical context learning, we introduce two types (patch-patch and patch-token) of hierarchical tracking maps (HTM) to characterize the patterns associated with different levels of image feature encoding. Patch-patch tracking maps seek to capture the mapping relationship between the visual context derived from the original image patches and the output image embeddings of ViT. On the other hand, patch-token tracking maps are employed to represent the mapping from original image patches to textual tokens. Using the HVFormer as a foundation, we empirically study the influence of hierarchical visual contexts through four perspectives: performance evaluation, the nature of auxiliary visual information, the patterns observed in the image encoding hierarchy, and the significance of various visual encoding levels. Extensive experiments performed on the MNRE benchmark dataset [45] demonstrate the effectiveness of HVFormer.

In summary, our main contributions are as follows:

- We propose a two-stage multimodal fusion model that first captures text-relevant visual information at the same level and subsequently leverages a mixture of multimodal experts to fuse visual features across different levels.
- We introduce the hierarchical tracking maps to characterize the patterns of visual semantics, spanning from local to global levels, as learned by different multimodal model layers, and we thoroughly analyze the implications of exploiting multi-level visual features for decoding relations.
- We conduct experiments on the multimodal relation extraction benchmark and compare them with state-of-the-art (SOTA) baselines. Empirical studies confirm the effectiveness of our proposed model.

## 2 RELATED WORKS

### 2.1 Multimodal Learning

Multimodal learning (MML) has been of significant importance in multimodal applications, dating back to some of the initial applications of audio-visual speech recognition in the 1980s. [40]. In recent years, Transformers have become the general architecture for MML tasks. Existing multimodal transformers can be summarized according to their network structures: 1) Models with a single-stream structure, including UNITER [6], VisualBERT [16], VL-BERT [27], and Unified VLP [46], combine the input from different modalities and feed them into the single model. 2) Models with multi-stream structures, such as ViLBERT [20], LXMERT [3], and ActBERT [47], separately process unimodal input in different streams with cross-modality structures. Our work follows the two-stream multimodal model, the most commonly adopted paradigm in multimodal information extraction.

### 2.2 Multimodal Relation Extraction

As a crucial component of information extraction, traditional relation extraction methods typically identify relationships in the textual model. However, texts on social media are often short, ambiguous, and lacking context. Zheng et al. [45] proposed that the visual content of the posts can supplement semantics and developed the first multimodal RE dataset. Previous work for MRE [32, 35, 44] mainly considers learning multimodal representation based on the final output of image encoders. The recent MKGformer [4] and IFAformer [43] utilize dual-modal alignments at the same layer of text and image encoders to implicitly exploit hierarchical visual context. HVPNeT [5] extracts coarse-grained vision embeddings relevant to text from each ResNet block and considers the importance of different visual blocks. However, understanding the mechanisms and characteristics of hierarchical context learning involved in multimodal models remains a challenge that has yet to be addressed.

### 2.3 Mixture of Experts

A mixture of experts (MoE) combines the outputs of sub-models, referred to as "experts," using a weighting function known as the "router." MoE is first proposed as an ensemble procedure to assign input cases to one or a few experts [11] that generate predictive distributions. Eigen et al. [9] employ MoEs as basic building blocks

at each layer in a multi-layer network to improve model performance. Ma et al. [21] adapt MoE to multi-task learning by sharing expert networks across tasks. Shazeer et al. [26] introduce Sparsely-Gated MoE consisting of thousands of feed-forward sub-networks for language modeling and machine translation. V-MoE [15] utilizes MoE to achieve a sparsely activated vision transformer model. LIMoE [22] is a recent application of MoE in multimodal learning, in which all input text and image are assigned to partially activated modality-agnostic MLP experts. Unlike LIMoE [22], we implement each expert to fuse the text and visual semantics represented at a specific level.

## 3 METHOD

### 3.1 Problem Formulation

Traditionally, the task of relation extraction is to predict the relation $r$ between a subject entity $e_s$ and an object entity $e_o$ in a sentence $S = \{w_1, \ldots w_n\}$, where $n$ is the length of the sentence. In this paper, we deal with the multimodal relation extraction problem, where an image $I$ is paired with $S$ to provide additional context information to determine the relation $r$. Let $\mathcal{Y} = \{r_1, r_2, \ldots, r_R\}$ be a set of $R$ distinct relation types. An MRE model is expected to generate the probability of $P(r|S, I, e_s, e_o)$ for each $r \in \mathcal{Y}$, given the input text $S$ with entities $e_s, e_o$ and image $I$.

### 3.2 Textual Semantic Representation

We use pre-trained BERT [7], a commonly applied text encoder to extract textual semantic representations from the text. To format the BERT input sequence $T_S$, we follow the same process as in previous work [4, 44], tokenizing the sentence $S$ using the BERT Wordpiece tokenizer [34] and adding special tokens $\langle s \rangle, \langle /s \rangle, \langle o \rangle, \langle /o \rangle$ to indicate the start and end of the subject and object entities. Additionally, we align the tokenized sequence with the BERT encoding process by prepending the special token $[CLS]$ at the start of the sequence and padding the end of the sequence to the max sequence length using the special token $[PAD]$. Accordingly, the input sequence $T_S$ is formatted as follows:

$$T_S = \{[CLS], t_1, \ldots \langle s \rangle, t_{e_s}, \langle /s \rangle, \ldots$$
$$\langle o \rangle, t_{e_o}, \langle /o \rangle, \ldots, [PAD]\} \quad (1)$$

where $t_{e_s}$ and $t_{e_o}$ are tokenized sub-sequences of the subject and object entities, respectively. BERT then processes each token $t_i \in T_S$ as a contextual vector representation. Formally, the textual semantic representation for the input sequence $T_S$ can be expressed as:

$$T = \text{BERT}(T_S) \quad (2)$$

where $T \in \mathbb{R}^{n_T \times d_T}$. $n_T$ is the sequence length and $d_T$ is the embedding dimension.

### 3.3 Visual Semantic Representation

Multimodal information extraction models typically acquire visual features from images by directly adopting pre-trained computer vision approaches [4, 5, 38]. CNN-based image encoders such as Faster R-CNN [25], ResNet [10], and transformer-based methods such as ViT [8] have been used in previous work [35, 38, 39]. Due to the recent success of ViT in multimodal information extraction [4, 35], we choose ViT as the visual backbone. The input image $I$ is

first rescaled to a specific dimension and then divided into non-overlapping 2D patches $V_S = \{i_1, i_2, \ldots, i_{n_V}\}$. A special class patch $[CLS]$ is prepended to the sequence $V_S$ to represent the image $I$. Each patch is then mapped to a one-dimensional embedding to generate the patch embedding matrix $V^0 \in \mathbb{R}^{n_V \times d_V}$, where $d_V$ is the embedding dimension. ViT processes the patch embedding matrix $V^0$ through multiple pre-trained transformer layers to obtain the visual semantic representation:

$$V^l = \text{ViT}(V^{l-1}), \; l \in \{1, 2, \ldots, L\} \quad (3)$$

where the $V^l \in \mathbb{R}^{n_V \times d_V}$ denotes the output embedding matrix of the $l$-th layer ViT and $L$ is the maximum layer.

### 3.4 Hierarchical Visual Context Fusion Transformer

Here, we develop a two-stage hierarchical visual context fusion transformer for MRE. Our model, depicted in Figure 2, comprises independent expert networks dedicated to the intra-level aggregation of token-aware visual semantic representations, and a shared expert router to facilitate the inter-level aggregation of multimodal representations preserving hierarchical visual features. In the following sections, we provide details of our approach.

*3.4.1 Visual Context Learner.* Our method employs textual-guided visual attention blocks (TVA) to learn complementary image features encoded at a specific level $l$. The approach is inspired by co-attentional transformer blocks, a common technique used in multimodal learning and multimodal information extraction [27, 36, 38] to combine the features from different modalities. The block uses the textual semantic representation $T \in \mathbb{R}^{n_T \times d_T}$ as queries and the visual semantic representation $V^l \in \mathbb{R}^{n_V \times d_V}$ extracted from the ViT layer $l$ as key-value pairs, thus allowing the block to learn token-aware visual representations. Formally, the cross-modal attention block operates as follows:

$$A_C^{(l,m)} = \text{Softmax}\left(\frac{[TW_q^m][V^l W_k^m]^T}{\sqrt{d/M}}\right)$$
$$head_m = A_C^{(l,m)}[V^l W_v^m] \quad (4)$$
$$\hat{V}_l = [head_1; head_2; \ldots; head_M]W_{att}$$

where $\{W_q^m, W_k^m, W_v^m\} \in \mathbb{R}^{d \times d/M}$ are the weight matrices specific to $head_i$ for generating key, query and value vectors. In our implementation, we let token embeddings have the same dimension as patch embeddings, so we use $d$ to represent the hidden dimension size of the unified dimension. $A_C^{(l,m)} \in \mathbb{R}^{n_T \times n_V}$ denotes the cross-modal attention map. $W_{att}$ is the weight matrix for $M$ attention heads. Our cross-modal attention block then incorporates the technique of layer normalization [1]:

$$\tilde{V}_l = \text{LayerNorm}(\hat{V}_l) \quad (5)$$

To fuse textual and visual representations, we modify the fully connected feed-forward network in vanilla Transformer [31]. The multimodal representations $C_l \in \mathbb{R}^{n_T \times d}$ are calculated as:

$$C_l = FFN_l([T; \tilde{V}_l]) + T$$
$$= h([T; \tilde{V}_l]W_1^l + b_1)W_2^l + b_2^l + T \quad (6)$$

**Figure 2: The macro-structure of our method for multimodal relation extraction.**

where $W_1 \in \mathbb{R}^{2d \times d}$, $W_2 \in \mathbb{R}^{d \times d}$ are the transformation matrices and $h(\cdot)$ is the activation function used in BERT.

*3.4.2 Mixture of Multimodal Experts.* We then introduce a mixture of multimodal experts (MoME) to explore whether distinct textual tokens may favor image information encoded at different levels. A mixture of experts combines the outputs of sub-models known as "experts" via a weighting function known as the "router" in an input-dependent way. The proposed mixture of multimodal experts is formulated as follows.

$$MoME(V, T) = \sum_{e=1}^{E} g(V, T)_e \cdot f_e(V, T) \quad (7)$$

where $f_e(\cdot)$ denotes the $e$-th expert function and $g(\cdot)$ is the routing function that prescribes the weights for the $E$ experts.

The visual context learner described in Section 3.4.1 is formally defined as the expert network $f_e(\cdot)$. We apply expert networks $\{f_0(V^0, T), f_1(V^1, T), f_2(V^2, T), \ldots, f_L(V^L, T)\}$ to different ViT layers, thus generating a hierarchical visual features set $\tilde{V}_H$ and a multimodal representations set $C_H$:

$$\tilde{V}_H = \{\tilde{V}_0, \tilde{V}_1, \tilde{V}_2, \ldots, \tilde{V}_L\}$$
$$C_H = \{C_0, C_1, C_2, \ldots, C_L\} \quad (8)$$

All visual embeddings in $\tilde{V}$ are concatenated and then fed into the expert router, which is a linear transformation of the input followed by a softmax layer:

$$G = g(\tilde{V}_H, T) = \text{Softmax}([T; \tilde{V}_0; \tilde{V}_l; \ldots; \tilde{V}_L] W_g) \quad (9)$$

where $W_g \in \mathbb{R}^{((L+2) \times d) \times E}$ is the trainable projection matrix and the output $G \in \mathbb{R}^{n_T \times E}$ preserves the predicted expert weights for diversed textual tokens.

Some previous studies [28, 38] on multimodal information extraction have highlighted the potential issue of noise within images. To handle the possible misalignment between text and visual data, we incorporate a specialized expert, $f_{text}(T) = T$, to provide text-only information. The introduction of $f_{text}(\cdot)$ serves to examine the necessity of integrating image semantics into the MRE task. Therefore, the number of experts $E$ equals $L + 2$. The final cross-modal semantic representations are calculated as follows:

$$C = \sum_{l=0}^{L} g(\tilde{V}_H, T)_l \cdot C_l + g(\tilde{V}_H, T)_{text} \cdot f_{text}(T)$$
$$= \sum_{l=0}^{L} G_{[:, l+1]} \cdot C_l + G_{[:, 0]} \cdot T \quad (10)$$

*3.4.3 Decoder.* We send the cross-modal representations to a linear projection layer with the softmax function to compute the predicted relation probability:

$$P_r = \text{Softmax}([C_{[]}; C_{[<o>]}] W_r^C) \quad (11)$$

where $W_r^C \in \mathbb{R}^{2d \times R}$ is the trainable projection matrix. $C_{[]}$ and $C_{[<o>]}$ are the representations of special tokens $\langle s \rangle$ and $\langle o \rangle$. The model parameters are optimized by minimizing the cross-entropy error between $P_r$ and the ground truth distribution:

$$\mathcal{L}_{re} = - \sum_{r \in \mathcal{Y}_r} y_r \cdot \log(p(r)) \quad (12)$$

## 3.5 Hierarchical Pattern Tracking Maps

To gain a reasonable understanding of the manner in hierarchical visual contexts are represented and aggregated, it is imperative to analyze the intrinsic mechanisms of image information processing within the multimodal model. In this paper, we propose hierarchical tracking maps (HTM) to characterize patterns associated with different image feature encoding levels. Given the input with $n_T$ text tokens and $n_V$ image patches, two types of multidimensional arrays

are calculated: the patch-patch tracking maps $H_V \in \mathbb{R}^{L \times n_V \times n_V}$ and the patch-token tracking maps $H_C \in \mathbb{R}^{L \times M \times n_T \times n_V}$. $H_V$ is leveraged to capture the mapping relationship between visual information derived from original image patches and image embeddings generated by ViT. $H_C$ is employed to represent the mapping from the initial image patches to the textual tokens. Figure 3 illustrates the computation process.

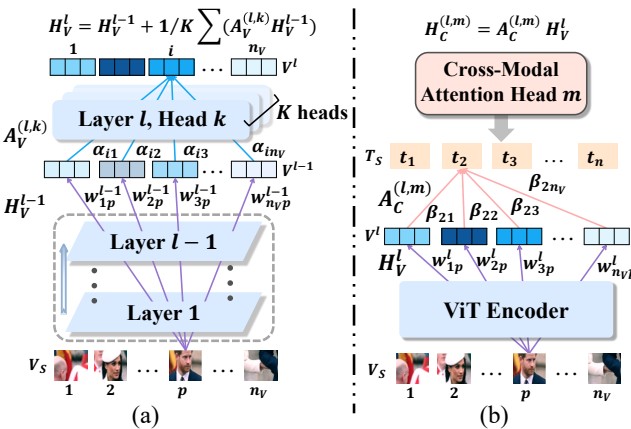

**Figure 3: (a) The process of calculating hierarchical patch-patch tracking maps $H_V^l$. (b) Inferring tracking maps $H_C^l$ between input image patches and textual tokens.**

The element $w_{ip}^l$ in $H_V^l \in \mathbb{R}^{n_V \times n_V}$ denotes the estimated importance weight of visual semantics from the original image patch $p$ in generating the image embedding of patch $i$ after $l$ ViT layers. $H_V^0$ signifies the initial tracking maps before any transformer blocks are applied, which is an identity matrix $I$ indicating that each patch exclusively accumulates information from its own context:

$$H_V^l = \begin{bmatrix} w_{11}^l & w_{12}^l & \cdots & w_{1n_V}^l \\ w_{21}^l & w_{22}^l & \cdots & w_{2n_V}^l \\ \vdots & \vdots & \ddots & \vdots \\ w_{n_V 1}^l & w_{n_V 2}^l & \cdots & w_{n_V n_V}^l \end{bmatrix}, H_V^0 = \begin{bmatrix} 1 & 0 & \cdots & 0 \\ 0 & 1 & \cdots & 0 \\ \vdots & \vdots & \ddots & \vdots \\ 0 & 0 & \cdots & 1 \end{bmatrix} \quad (13)$$

To infer the tracking maps $H_V$, it is essential to examine the computational structures within ViT. The self-attention maps computed in each ViT layer can serve as empirical evidence to indicate how patch embeddings are aggregated to produce new visual representations. Assuming that there are $K$ heads in the $l$-th ViT layer, we can extract the visual self-attention maps $A_V^l \in \mathbb{R}^{K \times n_V \times n_V}$. The vector $\alpha_i \in \mathbb{R}^{1 \times n_V}$ in the $i$-th row of the attention map $A_V^{(l,k)} \in \mathbb{R}^{n_V \times n_V}$ denotes the attention weights for the image patch $i$ in head $k$. These weights signify the importance coefficients of patch embeddings $V^{l-1}$ from the $(l-1)$-th ViT layer to obtain a new visual representation for patch $i$.

$$A_V^{(l,k)} = \begin{bmatrix} \alpha_{11} & \alpha_{12} & \cdots & \alpha_{1n_V} \\ \alpha_{21} & \alpha_{22} & \cdots & \alpha_{2n_V} \\ \vdots & \vdots & \ddots & \vdots \\ \alpha_{n_V 1} & \alpha_{n_V 2} & \cdots & \alpha_{n_V n_V} \end{bmatrix}, \sum_{j=1}^{n_V} \alpha_{ij} = 1 \quad (14)$$

Given that the image embeddings $V^{l-1}$ have already retained the original visual information, the hierarchical mapping matrix of $(l-1)$-th layer is $H_V^{l-1}$. We can approximate the significance of the visual context of the original image patch $p$ in the creation of the patch embedding $i$ as follows:

$$w_{ip}^l = \alpha_{i1} w_{1p}^{l-1} + \alpha_{i2} w_{2p}^{l-1} + \cdots + \alpha_{in_V} w_{n_V p}^{l-1} = \sum_{j=1}^{n_V} \alpha_{ij} w_{jp}^{l-1} \quad (15)$$

We further take into account the influence of residual connection operation implemented in self-attention blocks:

$$w_{ip}^l = w_{ip}^{l-1} + \sum_{j=1}^{n_V} \alpha_{ij} w_{jp}^{l-1} \quad (16)$$

As there are $K$ distinct attention heads in each ViT layer, we proceed to perform iterative multiplication between $H_V$ and $A_V^{(l,k)}$, followed by summation and normalization to yield the evaluated matrix $H_V^l$. The corresponding matrix formulation can be expressed as:

$$H_V^l = Normalization(H_V^{l-1} + \frac{1}{K}\sum_{k}^{K} A_V^{(l,k)} H_V^{l-1}) \quad (17)$$

We iteratively apply the Equation 17 from $l = 1$ to $l = L$ to obtain the complete maps $H_V$.

Since image embeddings from ViT layers ($l \geq 1$) have preserved the information from all image patches, the cross-modal attention maps $A_C^l$ are not suitable for accurately estimating how visual features from raw image patches is exploited to supplement additional context for text tokens. To this end, we infer the hierarchical patch-token tracking map $H_C^{(l,m)} \in \mathbb{R}^{n_T \times n_V}$ for each cross-modal attention head. This matrix maintains the estimated importance weights of the original visual features of input image patches, contributing to the provision of relevant contexts for textual semantic representations. By combining the cross-modal attention $A_C^l$ and hierarchical patch-patch tracking maps $H_V^l$, we get the maps $H_C^l$:

$$\begin{aligned} H_C^l &= [H_C^{(l,1)}, H_C^{(l,2)}, \dots, H_C^{(l,M)}] \\ H_C^{(l,m)} &= Normalization(A_C^{(l,m)} H_V^l) \end{aligned} \quad (18)$$

Then $H_C^l$ is calculated for all ViT layers and concatenated to obtain the final hierarchical patch-token tracking maps $H_C$.

## 4 EXPERIMENTS AND RESULTS

### 4.1 Experimental Settings

*4.1.1 Dataset.* To evaluate the performance of our approach for the MRE task, we adopt the most widely used multimodal neural relationship extraction dataset (MNRE) [45]. The statistics of MNRE are shown in Table 1. Each sample in MNRE includes the textual content of a post and its corresponding image crawled from Twitter. Given an image, we follow [42] and adopt the visual grounding toolkit [37] to obtain visual objects with the highest $t$ salience.

*4.1.2 Implementation Details.* We conduct experiments using ViT-B/32 as the visual backbone and pre-trained BERT-base as the textual encoder. Both models have a similar architecture, with a hidden dimension of 768, 12 attention heads and 12 layers. We practically find that applying individual expert networks for all ViT layers

leads to numerous trainable parameters and results in inadequate training of model parameters. Considering that the difference in visual contexts learned by adjacent layers may not be significant, we extract image patch representations from ViT layers 0, 6, and 12 to represent the visual information encoded at the lowest, intermediate and highest levels. We ran all the experiments three times with random seeds. More details are presented in the Appendix A.1.

| #Ent. | #Rel. | #Train | #Valid | #Test |
|---|---|---|---|---|
| 30,970 | 23 | 12,247 | 1,624 | 1,614 |

**Table 1: The statistics of MNRE dataset.**

*4.1.3 Baselines.* We compare our approach with current competitive approaches to demonstrate the superiority of our method. First, we choose conventional text-based models, including **PCNN** [41], **MTB** [2], and **BERT** [7]. We then compare our method with previous multimodal approaches as follows: 1) **VisualBERT** [16] is a pre-trained visual-language model with a single-stream structure. 2) **BERT+SG** [44] concatenates the textual representations of BERT and the visual features generated by the scene graph tool [29]. 3) **MEGA** [44] employs the graph alignment algorithm to match the textual entities and visual objects detected by the scene graph generation tool [29]. 4) **Xu et al** [35] propose a data discriminator that divides social media posts into a multimodal and a unimodal set. 5) **MKGformer** [4] is a new SOTA method that proposes a hybrid transformer to integrate visual and text representations. 6) **IFAformer** [43] aligns multimodal features between texts and images by utilizing textual and visual prefix-based attention. MKGformer and IFAformer implicitly harness hierarchical visual contexts by fusing visual and textual representations at the same layers of BERT and ViT. Our HVFormer also utilizes BERT and ViT as the encoders, while we explicitly incorporate multi-level visual features acquired from various ViT layers into the textual representations. 7) **HVPNeT** [5] utilizes multi-scaled visual features derived from 4 ResNet blocks to create informative multimodal representations. HVPNet extracts fixed hierarchical visual representations for all textual tokens and focuses on inter-level aggregation. In contrast, our method additionally incorporates intra-level aggregation to learn more refined token-aware hierarchical visual representations.

## 4.2 Performance Comparison

*4.2.1 Main Results.* We report the precision (P), recall (R), and F1 score (F1) achieved by each compared method on the MNRE. Table 2 summarizes the experimental results of the baselines and our proposed method. We present the average results and the standard deviations. Our approach outperforms all baselines, demonstrating its overall superior performance. Specifically, compared to the recent SOTA model MKGformer, which has the same text encoder and image encoder as our method, our model achieves relative increases of 1.75% F1 score, 1.72% recall score, and 1.77% precision score. Furthermore, our HVFormer exhibits superior performance compared to HVPNeT that also explicitly exploits visual features learned by different image encoder blocks, highlighting the effectiveness of our model in learning fine-grained hierarchical visual contexts.

| Modal | Methods | P | R | F1 |
|---|---|---|---|---|
| Text | PCNN | 62.85 | 49.69 | 55.49 |
| | MTB | 64.46 | 57.81 | 60.86 |
| | BERT | 61.89 | 66.09 | 63.86 |
| Text + Image | VisualBERT | 57.15 | 59.48 | 58.30 |
| | BERT+SG | 62.95 | 62.65 | 62.8 |
| | MEGA | 64.51 | 68.44 | 66.41 |
| | Xu et al. | 66.83 | 65.47 | 66.14 |
| | MKGformer | 82.67 | 81.25 | 81.95 |
| | IFAformer | 82.59 | 80.78 | 81.67 |
| | HVPNeT | 83.64 | 80.78 | 81.85 |
| | Ours | **84.14**±.49 | **82.65**±.33 | **83.39**±.23 |
| | *w/o* HVF | 82.58±.53 | 82.22±.41 | 82.40±.65 |
| | *w/o* TVA | 81.81±1.8 | 76.35±3.4 | 78.95±2.3 |
| | *w/o* MoME | 83.49±.59 | 81.66±.53 | 82.56±.54 |

**Table 2: Performance comparison of previous SOTA baseline models for multimodal RE on MNRE dataset.**

*4.2.2 Model Ablation.* To investigate the effectiveness of key components in our model, we conduct an ablation study using the following model variants: 1) *w/o* HVF indicates that only the final output visual representations of the ViT are utilized for MRE. 2) *w/o* TVA refers to the model removing the cross-modal attention block. We simply leverage the $[CLS]$ patch embeddings from different ViT layers as learned hierarchical visual features. 3) *w/o* MoME indicates that we concatenate $T$, $C_H$ and feed them into a general feed-forward network for fusion. As shown in Table 2, removing hierarchical visual features (*w/o* HVF) leads to a degradation in model performance, demonstrating that just exploiting the final visual representations is sub-optimal for MRE. *w/o* TVA drops 4.44 F1 score, indicating that cross-modal attention blocks are crucial in capturing fine-grained visual features. The model without the mixture of multimodal experts (*w/o* MoME) drops 0.83 F1 score, suggesting that estimating the importance of different levels of image encoding helps to learn more effective multimodal representations.

## 4.3 Analysis and Discussion

**RQ1**: *How does performance vary when leveraging visual semantics from distinct encoding levels?*

**A**: We evaluate the performance of model variants that solely implement one specific expert network $f_l(\cdot)$. As depicted in Figure 4, leveraging the initial patch embeddings $V^0$ to gain contextual visual cues leads to the lowest F1 score. The result suggests that stacking ViT encoding layers can improve the acquisition of complementary visual information. In addition, Figure 4 reveals that the use of high-level visual representations ($l = 11, 12$) can result in a performance decline to some extent while exploiting image embeddings derived from intermediate layers of ViT ($l = 5 - 7$) yields the optimal performance. We conjecture that this is primarily due to patch embeddings in high-level layers of ViT being generated by more aggregation operations, which makes it challenging to precisely identify relevant visual information.

**RQ2**: *What are the characteristics of the complementary visual contexts obtained from different layers of ViT?*

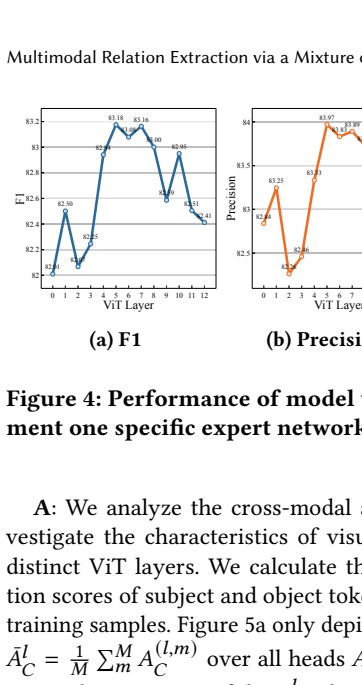

**(a) F1**     **(b) Precision**     **(c) Recall**

**Figure 4: Performance of model variants that solely implement one specific expert network $f_l(\cdot)$.**

**A**: We analyze the cross-modal attention distributions to investigate the characteristics of visual semantics captured from distinct ViT layers. We calculate the overall cross-modal attention scores of subject and object tokens ($\langle s \rangle, \langle o \rangle$) across all MNRE training samples. Figure 5a only depicts the average attention map $\bar{A}_C^l = \frac{1}{M} \sum_m^M A_C^{(l,m)}$ over all heads $A_C^{(l,m)}$ for space reason. For a comprehensive view of the $A_C^l$, please refer to the Appendix A.3.

We find that the kurtosis of cross-modal attention distribution is increased as image patches are processed by more vision transformer layers. Kurtosis is a statistical measure used to describe the "tailedness" of a probability distribution, which is computed as:

$$Kurtosis(P) = \frac{\mu_4}{\sigma^4} - 3 \qquad (19)$$

where $\mu_4$ is the central fourth moment and $\sigma$ denotes the standard deviation. The distribution with a high kurtosis typically has a heavy tail. The kurtosis of $A_C^l$ is calculated as the sum of the kurtosis values of all attention heads. As shown in Figure 5b, the $A_C^{12}$ has a much higher kurtosis (48.5) than $A_C^1$ (11.4). These results suggest that when leveraging visual embeddings learned by deeper ViT layers, the model generally concentrates on fewer patch embeddings to obtain complementary visual contexts. We speculate that a limited set of high-level patch embeddings may supply sufficient global visual information for relation identification.

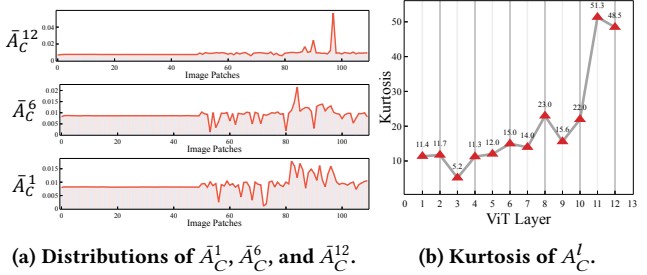

**(a) Distributions of $\bar{A}_C^1$, $\bar{A}_C^6$, and $\bar{A}_C^{12}$.**     **(b) Kurtosis of $A_C^l$.**

**Figure 5: Average cross-modal attention distribution $\bar{A}_C^l$ of subject and object tokens on MNRE train set.**

**RQ3**: *Can the hierarchical tracking maps be considered a reliable indicator for understanding the intrinsic mechanisms of image processing in multimodal models?*

**A**: We calculate the patch-patch tracking maps $H_V^l$ for all training samples, and Figure 6 presents the average results. It is observed that

$H_V^l$ of the earlier layer exhibits high scores on diagonal elements, suggesting that the queried image patches predominantly attend to their own local visual features. As layers go deeper, the significance of these local visual features gradually diminishes, and there is a shift towards focusing more on global information from other image patches. $H_V^l$ of intermediate ViT layers shows a unique pattern that balances weights on both local and global information. These observations provide insights into why optimal performance is achieved via utilizing mid-level patch embeddings to attain visual information, as reported in Figure 4.

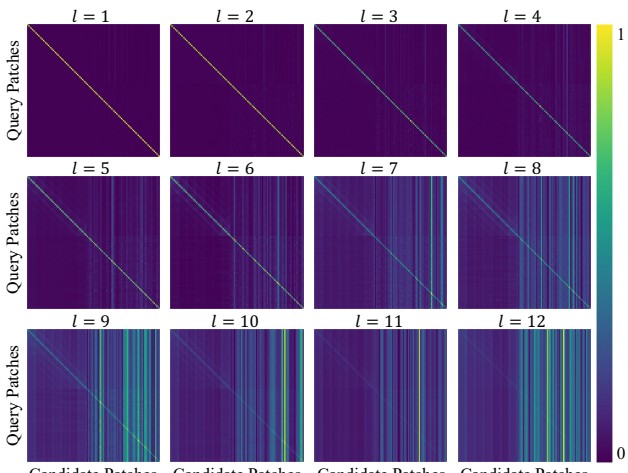

**Figure 6: Average hierarchical patch-patch tracking maps $H_V^l$ calculated by Equation 17. Best view in color.**

Moreover, it is noteworthy that the dynamic patterns of representing hierarchical visual context can affect the variation in the abilities of cross-modal attention heads to capture complementary visual contexts from input image patches. We compute the patch-token tracking maps $H_C^l$ for tokens $\langle s \rangle, \langle o \rangle$ in all training text posts and Figure 7a, b show visualizations of $H_C^1$ and $H_C^{12}$, respectively. To inspect the variation among patch-token matrix corresponding to different heads, we measure the KL-divergence between each $H_C^{(l,m)}$ and the averaged tracking matrix:

$$D_{KL}(H_C^l || \bar{H}_C^l) = \sum_{m=1}^{M} \sum_{i=1}^{n_V} M_i^{(l,m)} log\left(\frac{M_i^{(l,m)}}{\bar{M}_i^l}\right)$$

$$\bar{H}_C^l = \frac{1}{M} \sum_{m=1}^{M} H_C^{(l,m)} \qquad (20)$$

A high value of $D_{KL}(H_C^l || \bar{H}_C^l)$ suggests large disparities among the distributions of $H_C^{(l,m)}$. As illustrated in Figure 7c, the $D_{KL}$ value is reduced as the number of ViT layers increases. This observation can be attributed to high-level patch embeddings retaining more global visual information, consequently representing more consistent visual semantics.

Prior experiments predominantly utilize visual self-attention maps to probe the underlying mechanisms of learning hierarchical

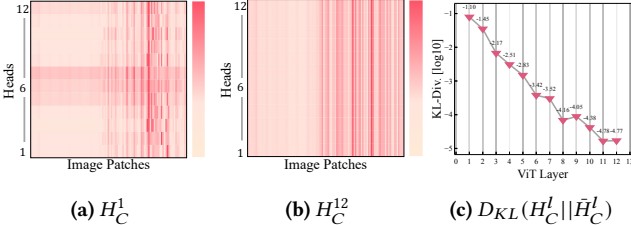

(a) $H_C^1$      (b) $H_C^{12}$      (c) $D_{KL}(H_C^l || \bar{H}_C^l)$

**Figure 7: Average hierarchical tracking maps $H_C^l$ between input image patches and textual tokens ($\langle s \rangle$, $\langle o \rangle$).**

visual contexts, which may raise concerns regarding the oversight of the effect of other structures in vision transformers (e.g., layer normalization, feed-forward network). To validate the analysis results above, we conduct a case study to visualize the contextual image features represented at different levels. We extract the output hidden patch representations of each ViT layer and employ t-SNE [30] to project these patch embeddings into a two-dimensional space, as shown in Figrue 8a. It is observed that as the layers go deeper, the spatial distance between the initial image embeddings ($l = 0$) and the hierarchical visual embeddings ($l \geq 1$) increases. The discrepancy between the visual representations of the entire image and those of the object sub-images generated by the same ViT layer gradually decreases. These phenomena further strengthen and align with our conclusions, as outlined above.

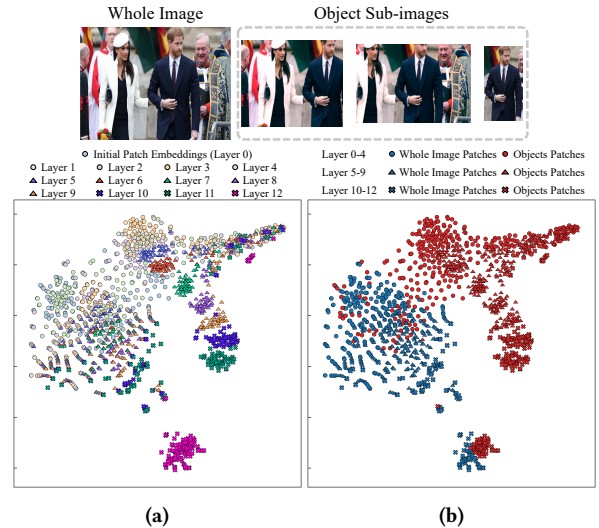

(a)          (b)

**Figure 8: (a) Visualization of all hierarchical visual representations. (b) Patch representations from the original whole image are marked in blue, while those from object sub-images are highlighted in red.**

**RQ4**: *What is the significance of different levels of vision encoding when using multiple-level visual features simultaneously?*

**A**: Expert weights reflect the significance of semantics derived from different visual encoding levels. Figure 9a reports the average expert weights for tokens $\langle s \rangle$ and $\langle o \rangle$ during the training process.

It can be seen that the weight of the text-only expert gradually decreases to zero, and the importance weights are mainly concentrated in experts fusing visual features from ViT layers 6 and 12. Figure 9b presents the overall distribution of expert weights when the model reaches convergence. These findings suggest that images provide critical contexts for decoding relations.

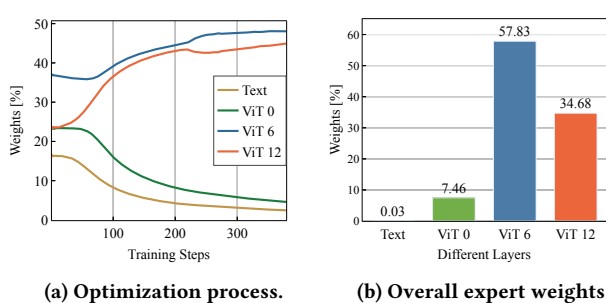

(a) **Optimization process.**      (b) **Overall expert weights.**

**Figure 9: The estimated importance weights for multimodal experts on MNRE train set.**

We summarize the expert weight distributions of subject and object tokens for different relations. Figure 10 presents the statistical distributions of expert weights for 4 relations on the MNRE test set. We observe that tokens with the same relation tend to rely on a primary expert, and classifying different relations may attend to visual features at different levels. For instance, the performance of predicting relation */member_of* relies on information from ViT layer 6, while inferring relation */part_of* focuses on visual context from layer 12. The results reveal that text-image pairs related to distinct relations may exhibit specific relational patterns in utilizing visual information at different levels.

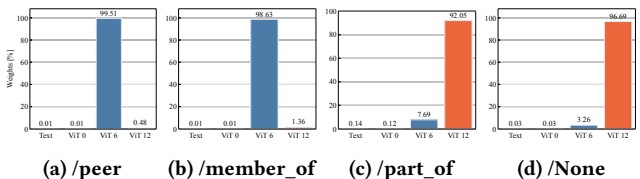

(a) **/peer**    (b) **/member_of**    (c) **/part_of**    (d) **/None**

**Figure 10: Expert weights of four relations: */per/per/peer*, */per/org/member_of*, */misc/misc/part_of* and */None*.**

## 5 CONCLUSION

In this paper, we present a hierarchical visual context fusion transformer for multimodal relation extraction. Our approach utilizes independent expert networks at each vision transformer layer to capture supporting visual cues from distinct image encoding levels. The textual and corresponding text-aware visual representations are then adaptively aggregated using importance weights predicted by an expert router. Hierarchical tracking maps are introduced to characterize the underlying patterns of image information processing within multimodal models. Extensive experiments are conducted to study how hierarchical visual contexts affect relation extraction. Our experimental results demonstrate that the proposed method outperforms existing state-of-the-art approaches.

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

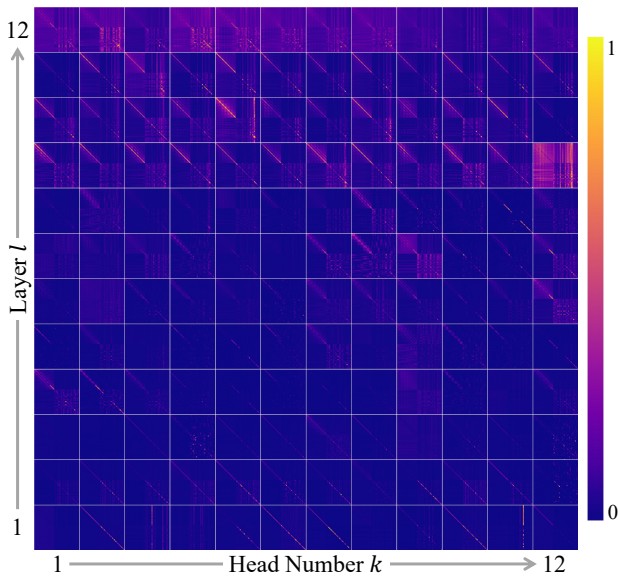

**Figure 11: The average self-attention map $A_V^l$ of ViT on MNRE train set. Best view in color.**

# A APPENDIX

## A.1 Implementation Details

We conduct experiments using ViT-B/32 as the visual backbone and pre-trained BERT-base as the textual encoder. Both pre-trained BERT and ViT are obtained from the HuggingFace Transformer repository.[2] Both models have a similar architecture, with a hidden dimension of 768, 12 attention heads, and 12 layers. The textual-guided cross-modal attention block is implemented with $M = 12$ heads.

We optimize the model using AdamW [19] with an initial learning rate of $lr = 1 \times 10^{-5}$ over 20 epochs. We implement our code using the PyTorch deep learning framework [23] and run all experiments on NVIDIA Tesla V100 GPU. It takes nearly 6 hours to train the model in our experimental environment. To reduce random bias, we run all experiments three times and report the average results as well as the standard deviations. The source code of our model will be made publicly available after the review process if accepted.

Applying individual expert networks for all image encoding levels leads to numerous trainable parameters. We practically find that such an implementation can result in inadequate training of expert-specific parameters. Considering that the difference in visual context learned by adjacent layers may not be significant, we extract image patch representations from ViT layers 0, 6, and 12 to represent the visual information encoded in the lowest, intermediate, and highest levels. To this end, the total number of parameters in our model is 250.8M.

---

[2]https://github.com/huggingface/transformers

## A.2 Self-attention Maps of ViT

Figure 11 shows the average visual attention map of ViT on the MNRE train set. It can be observed that attention heads in earlier layers (e.g., $A_V^1$) yield high activation scores on diagonal elements, meaning that query image patches focus on local information from themselves. As layer depth increases, certain attention heads exhibit varied patterns where scores are distributed over a wide range of image patches rather than being concentrated in diagonal elements. This observation aligns with findings from prior research [14]. Such inherent characteristics of ViT in image information encoding suggest that the output visual embeddings from deeper ViT layers are generated by preserving more global visual features.

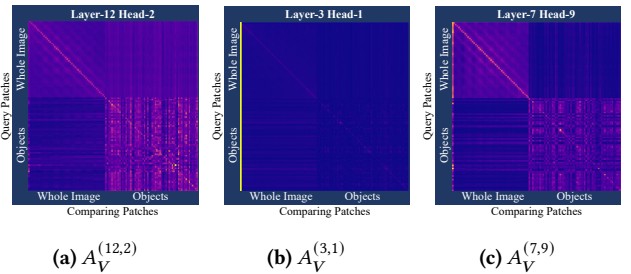

**(a)** $A_V^{(12,2)}$     **(b)** $A_V^{(3,1)}$     **(c)** $A_V^{(7,9)}$

**Figure 12: (a) An example of heads acquiring global visual information from all input image patches. (b) An example of heads attending to $[CLS]$ patch that represents the entire image. (c) $A_V^{(7,9)}$ aggregates global visual information from both sides.**

We further notice that global visual information can be obtained in different ways. Figure 12a presents an example of heads aggregating visual features from all image patches. Note that since both the original complete images and their corresponding object sub-images are input to the vision encoder, $A_V^{(12,2)}$ is partitioned into four blocks with distinct patterns. Queries from the original image have different association patterns regarding patches from the original image and those from object images. Figure 12b shows that the head $A_V^{(3,1)}$ employs an alternative approach to capture global visual information. All queries in head $A_V^{(3,1)}$ have exceptionally high activation scores to the special class patch $[CLS]$ that represents the semantics of the entire image. Actually, more attention heads gather visual information from both two sides, as illustrated in Figure 12c.

## A.3 Cross-modal Attention Maps

To study the impact of visual encoding hierarchy in offering text-relevant context, we present the average cross-modal attention distributions $A_C^l$ for subject and object tokens in MNRE training samples, as illustrated in Figure 13. It is evident that distinct attention heads within the same layer $l$ can have distinctive distribution patterns. Some heads tend to assign uniform weights across all image patches (e.g., $A_C^{(10,12)}$), whereas others may display more concentrated distributions (e.g., $A_C^{(10,1)}$). Overall, as we exploit image embeddings produced by deeper ViT layers, the kurtosis of cross-modal attention distribution tends to rise, as demonstrated

in Figure 4b. Obviously, most attention heads of $A_C^{11}$ and $A_C^{12}$ have heavy tails and are concentrated around several image patches, which means that the model can acquire enough global image information from a limited set of high-level visual representations.

Received 20 February 2007; revised 12 March 2009; accepted 5 June 2009

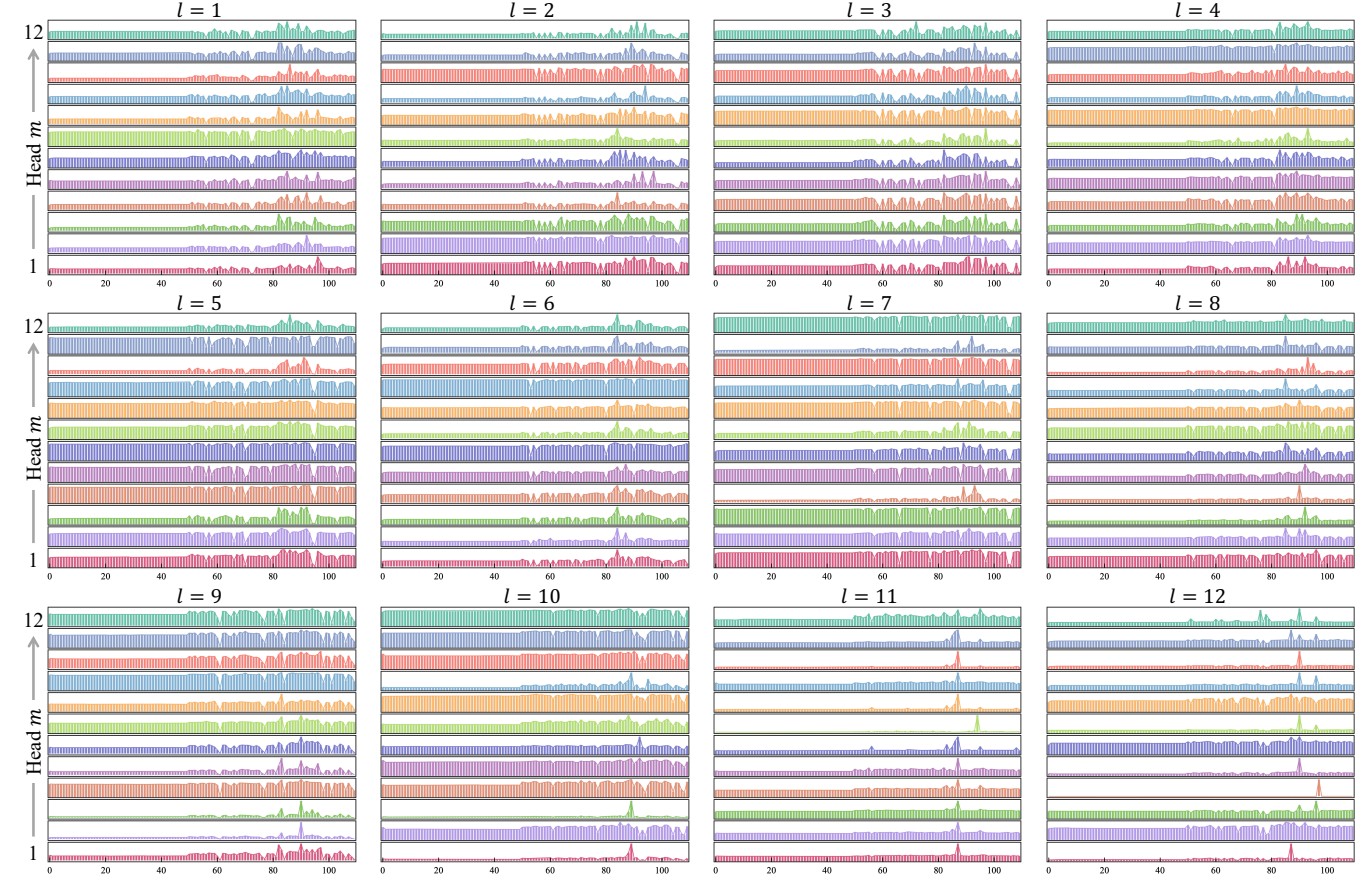

**Figure 13: The average cross-attention map $A_C^{(l,m)}$ on MNRE train set.**

