# OpenReview forum: "Multimodal Relation Extraction via a Mixture of Hierarchical Visual Context Learners"
_ACM.org/TheWebConf/2024/Conference — TheWebConf24_

### Official Review · Reviewer_P4kp · 2023-11-12

**Novelty:** 4
**Technical Quality:** 4

**Review:**

The paper proposes a novel approach to multimodal relation extraction by integrating hierarchical visual features into textual semantic representations. This is achieved through a two-stage hierarchical visual context fusion transformer paired with a mixture of multimodal experts framework. The concept of hierarchical tracking maps is introduced, aiming to enhance the understanding of image information processing within multimodal models. The paper claims state-of-the-art performance on the MNRE dataset.

**Strengths:**
1. The idea of the mixture of hierarchical visual context is interesting.
2. The experimental results consistently demonstrate performance improvements across many baselines on MNRE.
3. The paper is well-written and easy to follow.

**Weaknesses:**

1. The generalization of the methods deserves more testing. In this paper, methods only tested on the Multimodal Relation Extraction task(MRE), while many papers published on top-conference like [1][2][3] are tested on both MRE and Multimodal NER.
2. It's unclear what other expenses are linked with operating in this work.
3. More information on the nature and diversity of the auxiliary visual information used would be beneficial.
4. The paper could benefit from a discussion on the model's limitations and potential areas for improvement.

[1] Prompt Me Up: Unleashing the Power of Alignments for
Multimodal Entity and Relation Extraction. In MM. 2023.
[2] Hybrid Transformer with Multi-level Fusion for
Multimodal Knowledge Graph Completion. In SIGIR. 2022.
[3] Grounded Multimodal Named Entity Recognition on Social Media. In NAACL. 2022.

**Questions:**

Please see above.

**Reviewer Confidence:**

3: The reviewer is confident but not certain that the evaluation is correct

**Scope:**

3: The work is somewhat relevant to the Web and to the track, and is of narrow interest to a sub-community

---

### Official Review · Reviewer_C4br · 2023-11-14

**Novelty:** 5
**Technical Quality:** 6

**Review:**

The authors propose a novel two-stage hierarchical visual context fusion transformer incorporating the mixture of multimodal experts framework to effectively represent and integrate hierarchical visual features into textual semantic representations. The authors introduce the concept of hierarchical tracking maps to facilitate the understanding of the intrinsic mechanisms of image information processing involved in multimodal models.

Strong Points:
1. The research problem is very important. Multimodal relation extraction is important for web information mining.
2. The description of the model is very detailed. The article is well written.
3. The author's analysis and discussion of the experiment section is adequate.

**Questions:**

Weak Points:
1, The authors propose to utilize hierarchical visual information. But why not use hierarchical textual information?
2, The authors experimented on only one dataset. It is recommended that the authors experiment on multiple multimodal relationship extraction datasets to verify the validity of the model.

**Ethics Review Description:**

No.

**Reviewer Confidence:**

3: The reviewer is confident but not certain that the evaluation is correct

**Scope:**

4: The work is relevant to the Web and to the track, and is of broad interest to the community

---

### Official Review · Reviewer_ix3f · 2023-11-17

**Novelty:** 6
**Technical Quality:** 5

**Review:**

This paper addressed a web-related challenging issue which is to identify relationship between entities using multimodal input. The paper is well written. The proposed method, which deeply fuses two features from image and text encoders, respectively, by adding a cross-attention layer for each of ViT layer to pay cross attention to input text feature. Finally, a mixture-of-experts model is applied to fuse the deeply fused multimodal features which is fed to a classification head to identify relationships. It's innovative to leverage cross-attention for each and every layer of ViT and leverage MOE to refuse output of all layers. The experiments indicate this approach improves relationship extraction performance on a public dataset MNRE.

**Questions:**

The main concern is short of model performance evaluation. The proposed method is only evaluated on one dataset. Authors may consider other datasets, like the one in 'FL-MSRE: A Few-Shot Learning based Approach to Multimodal Social Relation Extraction'.  Additionally, it will make the paper more convincing by incorporate more ablation studies. For example, the proposed architecture leverage BERT as text encoder and ViT as visual encoder, and It would be convincing to provide model performance using other text encoder. The model provide the MOE weights of text feature and features from different fused ViT layers. It would be more convincing to show model performance on MNRE without text feature or with only a few of the fused ViT layers.

**Reviewer Confidence:**

3: The reviewer is confident but not certain that the evaluation is correct

**Scope:**

4: The work is relevant to the Web and to the track, and is of broad interest to the community

---

### Official Review · Reviewer_T59C · 2023-11-20

**Novelty:** 5
**Technical Quality:** 5

**Review:**

This paper introduces a new method for integrating hierarchical visual features into textual semantic representations, enhancing relation extraction effectiveness. It utilizes a two-stage hierarchical visual context fusion transformer along with a mixture of multimodal experts framework, aiming to boost performance and deepen the understanding of image information processing. The paper is well-structured and has clear motivations.

Reasons to accept:

1.The paper proposes a two-stage multimodal fusion model that initially captures text-relevant visual information, followed by employing multimodal experts to integrate visual features at various levels.

2.It presents hierarchical tracking maps to discern visual semantics from local to global scales within the multimodal model layers, using multi-level visual features to decode relationships.

3.The paper includes extensive experimental analysis, demonstrating the effectiveness of the proposed approach.

Reasons to reject:

1.Although Figure 2 intuitively explains the methodology, Section 3 describes it in an overly complex manner, making it difficult for readers to grasp. Additionally, some symbols are introduced without proper definition, complicating the understanding of the proposed approach.

2.Certain experimental results lack clarity and motivation. The rationale behind conducting specific analyses is not straightforward, and a more detailed case study with error analysis would enhance the paper's comprehensibility.

**Questions:**

See reasons to reject.

**Reviewer Confidence:**

3: The reviewer is confident but not certain that the evaluation is correct

**Scope:**

3: The work is somewhat relevant to the Web and to the track, and is of narrow interest to a sub-community

---

### Official Review · Reviewer_H68d · 2023-11-21

**Novelty:** 5
**Technical Quality:** 6

**Review:**

The paper deals with the problem of multimodal relation extraction and proposes a novel two-stage hierarchical visual context fusion transformer. The model incorporates a mixture of multimodal experts framework to represent and integrate hierarchical visual features into textual semantic representations.

The paper is well-written and all in all easy to follow, even for a non-expert (which is my case).

The related work is concise but well structured and the positioning of the paper wrt to previous approaches is clear within each of the three axes of work.

The method section would benefit from a running example (for instance one taken from the introduction figure 1).

The experiments are well-designed and the results are convincing. However, given the complexity of the architecture, I’d appreciate giving some detail regarding the efficiency of the approach as compared to the baselines, in particular given the only minor improvements over them.

Minor:
- The experiments are not a contribution, I’d remove them from the list in the introduction - they are means to demonstrate the quality / relevance of the contribution (which is the novel model)

**Questions:**

What is the computational efficiency of the new model wrt the baselines?

**Ethics Review Description:**

no issues

**Reviewer Confidence:**

2: The reviewer is willing to defend the evaluation, but it is likely that the reviewer did not understand parts of the paper

**Scope:**

4: The work is relevant to the Web and to the track, and is of broad interest to the community

---

### Decision · Program_Chairs · 2024-01-22

**Decision:**

Accept

**Comment:**

This paper tackles a challenging issue in the realm of web-related research, focusing on identifying relationships between entities using multimodal input. The paper is well-crafted, presenting a method that intricately fuses features from image and text encoders. This fusion involves incorporating a cross-attention layer for each ViT layer, enabling cross-attention to the input text feature. Ultimately, a mixture-of-experts (MOE) model is employed to amalgamate the deeply fused multimodal features, which are then fed into a classification head for relationship identification. The innovation lies in the utilization of cross-attention for every layer of ViT and leveraging MOE to refine the output of all layers. Experimental results indicate that this approach significantly enhances relationship extraction performance on the MNRE public dataset.

 All of the reviewers agree that this work is interesting with solid results. However, there are still a few weaknesses pointed out by the reviewers. The authors have addressed most of the concerns in the rebuttal. I would encourage the authors to carefully incorporate the discussions with the new results into their revision.